# Human–Fungal Pathogen Interactions from the Perspective of Immunoproteomics Analyses

**DOI:** 10.3390/ijms25063531

**Published:** 2024-03-20

**Authors:** Tanaporn Wangsanut, Monsicha Pongpom

**Affiliations:** Department of Microbiology, Faculty of Medicine, Chiang Mai University, Chiang Mai 50200, Thailand; monsicha.p@cmu.ac.th

**Keywords:** host–pathogen interaction, antigenic proteins, immunoproteomics, fungal pathogens, antibody-based therapies

## Abstract

Antibody immunity is now known to play a critical role in combating mycotic infections. The identification of molecules that can elicit an antibody response against fungal pathogens is the first step in developing antibody-based therapeutic strategies. Antigenic proteins are molecules recognized by the immune system that can stimulate antibody production and, therefore, can be a direct target for studying human–fungal pathogen interactions. Advances in recent immunoproteomic approaches have substantially aided in determining the key antigenic proteins on a large scale. In this review, we present a collection of antigenic proteins identified in yeast, dimorphic, and filamentous fungal pathogens to date. The general features of antigenic proteins are summarized and reveal that the proteins could commonly function in antistress responses, protein synthesis, and metabolism. The antigenic proteins listed here could serve as starting materials for developing species-specific or broad-spectrum diagnostic tests, therapeutic antibodies, and even vaccines against fungal infections.

## 1. Human Mycotic Diseases: The Need for Better Therapeutic Interventions

Human mycotic diseases have been expanding worldwide and have become a major public health problem [1]. Indeed, the global burden of fungal diseases is estimated to be over 5.7 billion [1]. The most at-risk people include those with immunocompromised systems, such as HIV/AIDS and cancer patients. In addition, individuals taking antibiotics or immune-suppressing drugs, undergoing invasive medical treatments, or being admitted to an intensive care unit (ICU) are more susceptible to fungal infections. In recent years, patients with the coronavirus disease (COVID-19) have been identified as a new group at risk of invasive fungal infections [2]. Furthermore, climate change is associated with the emergence of new fungal pathogens globally [3,4]. As a result, the numbers of invasive fungal diseases are rising while at-risk populations also continue to increase and emerge. To respond to this global threat caused by fungal pathogens, the World Health Organization (WHO) recently developed the WHO fungal priority pathogen list. This list includes 19 fungal pathogens that are the causative agents of acute and subacute systemic fungal infections, for which disease treatment and control are challenging issues. The recognition of fungal pathogens as a major contributor to mortality is an important step forward in reducing the incidence of serious fungal diseases.

High-quality fungal diagnostics and potent antifungal drugs are key to saving patients with mycotic diseases [1,5]. Despite the discovery of new drugs and the development of therapeutic methods, some fungal pathogens remain resistant to the available antifungal drugs [5,6,7]. Likewise, mycology diagnostic capacity is usually limited due to the lack of rapid, affordable, and accurate diagnostic tools. In many settings, especially in low-resource countries, diagnostic tests for specific fungal diseases are not even available in reference laboratories, resulting in undiagnosed and untreated fungal infection cases. Thus, the development of novel therapeutic strategies and fungal detection tools is urgently needed for the efficient management of fungal infections.

## 2. Antibodies Mediate the Host Defense against Fungi

Immunity plays an essential role in protecting humans from fungal diseases because invasive fungal diseases are rare in immunocompetent individuals. Of the estimated 12 million fungal species [8], only a small number are considered major human pathogens. True fungal pathogens can attack immunocompetent hosts, and very few fungal species (*Blastomyces dermatitidis*, *Coccidioides immitis*, *Paracoccidioides brasiliensis*, and *Histoplasma capsulatum*) belong to this group of pathogens. In contrast, opportunistic fungal pathogens employ the host-weakened immune system to spread infection and cause disease. The most common opportunistic fungal pathogens are *Aspergillus*, *Candida*, *Cryptococcus*, and *Pneumocystis*, accounting for 90% of fatal mycoses [9]. However, it is noted that nearly every fungus can become an opportunistic or accidental pathogen in a human host with severe immune system defects [10]. In addition, human mycoses can occur in individuals with disproportionally strong immune responses, such as in cases of allergic conditions [11]. Thus, the modulation of the human immune system can be one of the strategies to prevent fungal infections and treat mycotic diseases.

The human immune system is composed of two parts: innate immunity and adaptive immunity. Innate immunity is an antigen-independent defense mechanism, and it is not specific to invading pathogens. In contrast, adaptive immunity involves specific responses to pathogens, requiring antigen exposure and innate immunity. The adaptive immune system, in response to fungal invasion, has two major arms: (i) cellular-mediated immunity (CMI) and (ii) humoral-mediated immunity (HMI or antibody-mediated immunity). Several topics related to fungal immunity, such as cellular recognition [12,13], innate immunity [14,15], adaptive immunity [16,17], and predisposed genetic factors leading to fungal infections [18], have been analyzed in other outstanding reviews. In this review section, the humoral responses mediated by B-cells and antibodies are mainly discussed to emphasize the significance of identified antigenic proteins from pathogenic fungi.

While it is well documented that CMI is the primary arm for the host defense against fungi, the critical role of antibody function against medically important fungi has only recently emerged [18,19]. The evidence that B-cells and antibodies contribute to human fungal immunity comes from the identification of humoral defects that predispose humans to a spectrum of mycotic diseases (Table 1). First, for example, humans with X-linked hyper-IgM syndrome (XHIGM), characterized by a disorder of T-cell and B-cell functions, have increased susceptibility to many fungal infections [20]. Specifically, mutations in the CD40L gene affect signaling in B-cell activation and differentiation, leading to hyper-IgM syndrome. Indeed, CD40L mutations or XHIGM were confirmed in patients with disseminated talaromycosis [21], histoplasmosis [22], candidiasis [22], cryptococcosis [23,24,25], and *Pneumocystis jirovecii* infections [22,26,27].

Second, a deficiency in the caspase recruitment domain-containing protein 9 (CARD9), a central signaling molecule of innate and adaptive immunity, predisposes humans to severe fungal infections [30,40]. In the case of *Candida albicans*, a normal human microflora and a major opportunistic pathogen, CARD9 is required for the generation of antifungal antibodies that suppress the pathogenic lifestyle of this fungus [30]. In fact, the CARD9-deficient patients who developed systemic candidiasis produced significantly lower levels of IgG antibodies [31]. Additionally, human IgA antibodies promote *C. albicans* commensalism in mucosal niches (such as the gut and oral cavity) by preferentially targeting and inhibiting the pathogenic hyphal form of *C. albicans* [41,42]. Wich et al. demonstrated that human serum antibodies confer protection against invading *C. albicans* via the inhibition of adherence to, invasion into, and damage of oral epithelial cells [43]. These reports collectively indicate that CARD9-mediated antibody production can protect humans against disseminated *Candida* infection.

Third, individuals with hyper IgE syndromes, characterized by elevated serum IgE, are at high risk for several recurrent fungal infections [44]. Deficiency in STAT3, an important transcription factor of the immune regulatory pathway, is one of the most common causes of hyper IgE syndrome [32,45,46]. Invasive fungal diseases were reported in patients diagnosed with STAT3-mutated hyper-IgE syndrome. The fungi isolated from affected patients included *Candida* [33,34], *Aspergillus* [35,36,37], *Histoplasma* [35], *Talaromyces* [21], *Coccidioides* [38], *Cryptococcus* [35,38], and *Fusarium* species [39].

Together, studies on the inborn errors of immunity highlight the role of B-cells and antibodies in protecting humans against mycotic diseases (Table 1). Of course, impaired B-cell and antibody production may not be the only direct cause of increased susceptibility to fungal diseases in these patients. Readers who are interested in full details on the clinical features and immunological defects associated with primary immunodeficiency disorders discussed in this article should refer to other excellent reviews [18,47].

## 3. Discovery of Antigenic Proteins in Fungal Pathogens: The First Major Step Forward in Developing Antibody-Based Therapeutic Agents

Antibody-based diagnosis and therapy have become powerful tools in managing infectious diseases because they have high specificity to their targets. This is particularly useful for controlling mycotic infections because fungi and humans are eukaryotes that share high similarities in physiology. In addition, antifungal monoclonal antibodies can potentially modulate and enhance the host’s immune response to fungal pathogens. This is especially beneficial in treating immunocompromised patients who are more susceptible to fungal infections. For disease prevention, antibody-mediated immunity is fundamental for vaccinations that efficiently induce protection against many infectious diseases [48,49] and likely against mycotic infections.

### The Methods and the Challenges

Recently, so-called “omics” technologies (e.g., genomics, transcriptomics, metabolomics, etc.) enable investigators to discover a collection of biological molecules on a large scale. Immunoproteomics approaches are generally performed to identify the total pool of antigenic proteins (Table 2) [50]. In fungal pathogen studies, antibodies derived from the sera of either patients or immunized animals are generally used to screen the protein extracts derived from pathogens. The most common immuno-screening methods include serological proteome analysis (SERPA) and the serological analysis of recombinant cDNA expression libraries (SEREX). In the SERPA method, mass spectrometry analysis is used to identify the sequences of a protein or peptide, while DNA sequencing is used to identify unknown cDNA in the SEREX method. Most fungal antigenic proteins have been identified via mass spectrometry (the SERPA approach) (Table 2). On the other hand, the SEREX approach is less commonly used in the identification of fungal antigens; only one study in *T. marneffei* utilized this approach so far (Table 2; [51]). The immunogenic property is then confirmed via a Western blot assay using the purified recombinant protein version of the antigenic protein of interest with the antibodies derived from patients’ sera. In many studies, bioinformatics analyses were employed to predict the epitopes of antigens. Potentially, the predicted epitopes can be generated as synthetic peptides for the development of fungal diagnostic tools or for producing monoclonal antibodies. Indeed, synthetic peptides can be more rapidly generated than recombinant proteins for further downstream applications (Figure 1).

Importantly, there are several issues that can impact the results of the identified antigens, which means it is challenging to compare the information from each study (Figure 1). First, the cellular sources where the antigenic proteins were extracted are varied. Pathogenic antigens were traditionally thought to be restricted to secreted or cell surface proteins to react with the host antibodies [65]. As a result, many studies extracted the proteins solely from the cell wall or collected only secreted proteins for immunoproteomics experiments. On the other hand, some later studies utilized the whole-cell lysates in screening for the antigenic proteins and obtained exciting results that indicated intracellular proteins could stimulate antibody production (see next section). Second, the morphological forms or cell fates of fungi can affect the results of the identified antigenic proteins. The cellular components and metabolic programs are drastically different between each cell type. Many fungal pathogens have the ability to switch morphology between hyphal and yeast cells. Conidia, another developmental cell type, are considered as infectious asexual spores for many fungal pathogens. In certain types of fungi (such as thermal dimorphic fungi), the pathogenic form can be distinguished from the non-pathogenic form, while in others (such as *Aspergillus* or *C. albicans*), the disease-related form cannot be easily determined. For dimorphic fungi, the antigenic proteins have been identified only in the pathogenic yeast form (Table 2). In *A. fumigatus*, different key immunogenic proteins were discovered when different cell types were investigated in an immunoproteomics experiment. As shown in Table 2, the cytochrome p450 and translation elongation factor eEF-3 were identified as putative biomarkers when using *A. fumigatus* in germling cells, while the thioredoxin reductase GliT was identified as a potential biomarker when using *A. fumigatus* in the mold form [52,53]. Third, the difference in results could be from the various sources of the derived antibodies, which usually come from patients or immunized animal models of infection. For example, fourteen novel proteins were identified as immunoreactive proteins when the sera from Cryptococcal meningitis patients were used in the experiment instead of the sera from mice immunized against pulmonary *C. neoformans* infection [58,59]. Lastly, different fungal strains can exhibit different antigenic protein profiles. In the *Cryptococcus gattii* antigenic protein study, for instance, only 6 out of the 68 immunoreactive proteins were common to three different isolates of the *C. gattii* genotype VGII [57]. Likewise, the MPLP6 mannoprotein was discovered to be a yeast-specific antigenic protein from the *T. marneffei* F4 strain, while the MP1 mannoprotein was identified as an antigenic protein from the *T. marneffei* PM1 strain [66,67]. Together, these issues can lead to challenges in selecting potential targets for the development of species-specific and broad-spectrum antibody-based therapeutic agents.

## 4. Current Progress in the Identification of Fungal Antigenic Proteins

With current advances in immunoproteomic technologies, a plethora of information is available and awaiting to be functionally characterized. However, the big question is which antigenic molecules deserve additional follow-up? Historically, follow-up studies usually focus on a specific molecule of a single fungal pathogen, as illustrated in Table 2. This is beneficial in determining antigenic proteins for use as diagnostic markers and vaccine candidates for specific fungal pathogens. Accordingly, non-pathogen-specific antigens that are presented across different fungal species are often referred to as “cross-reactive”. At the functional level, the “cross-reactive” antigens can be referred to as “common” or “universal” targets for antibody immunity [58]. When considering vaccine development, protection against more than one pathogen with a single treatment could be beneficial to humans as it could minimize the dangers experienced by encountering each pathogen [68]. Thus, in this review, we emphasized the values of common antigenic proteins found in many fungal pathogens, as described below (Table 3). It is noteworthy to point out that, even though we categorized these antigenic proteins as “common” targets at functional levels, they could exhibit fungal-specific activity to the host immunity due to evolutionary divergence at protein structural levels. For instance, catalase from *H. capsulatum* exhibited a species-specific reaction despite being a common antioxidant with antigenic properties in multiple fungal species [61] (Table 3). Likewise, eEF-3 showed no homology with human proteins and little sequence homology with fungal proteins from the Mucorales order, *Penicillium* spp., *Paracoccidioides brasiliensis*, *Fusarium* spp., and *Paecilomyces* spp. [52], even though the translation process is a conserved pathway across eukaryotes. Hence, the antigenic proteins listed here could be exploited as both broad- and narrow-spectrum therapeutic targets.

As mentioned previously, immunoproteomic experiments with fungal whole-cell lysates revealed surprising results; intracellular proteins could be targets of AMI. Since surface or secreted proteins have been expected and known to function during host–pathogen interactions, these antigens were excluded from our analysis. The intracellular proteins with antigenic properties were our focus and are described below (Table 3).

### 4.1. Heat-Shock Proteins

The heat-shock proteins (Hsps) facilitate the proper folding and modification of proteins and are, therefore, involved in various biological functions during normal and unfavorable conditions [73]. In the context of human infection, fungal pathogens encounter several harsh environments in the host, including elevated host body temperature and the host immune response [73,74]. Hsps play an important role during host–pathogen interactions, and their roles in mediating virulence-related traits have been reported in many fungal studies [73,74]. Interestingly, various families of Hsps have been detected as antigenic proteins in many fungal pathogens, including *A. fumigatus* [52], *Candida* spp. [54,56,69], *Cryptococcus* spp. [57,58,60], *Paracoccidioides* spp. [62,70], and *T. marneffei* [64] (Table 3).

### 4.2. Carbon Metabolism

Carbon metabolism is pivotal for cellular function, and therefore, metabolic flexibility can enhance the fitness and pathogenicity of pathogenic fungi [75]. As glucose amounts can be limited in certain body sites, many fungal pathogens can assimilate to non-glucose substrates, which can be metabolized via gluconeogenesis and the glyoxylate cycle to generate hexose and pentose sugars [75,76]. The disruption of carbon metabolic functions can impair the growth and virulence of fungal pathogens, emphasizing the essential role of central carbon metabolism in fungal pathogenicity [75,77]. Importantly, the enzymes involved in glycolysis, gluconeogenesis, the electron transport chain, and the glyoxylate and TCA cycles have been identified as antigenic proteins in *A. fumigatus* [52], *C. albicans* [54,69,78], *C. parapsilosis* [56], *C. posadasii* [60], *C. gattii* [57], *C. neoformans* [58], *H. capsulatum* [61], *Paracoccidioides* spp. [62,70,71], and *T. marneffei* [64] (Table 3).

### 4.3. Translation

When organisms encounter stress, global translational repression allows the cells to reallocate valuable energetic resources to mount an appropriate response to specific environmental stressors [79,80]. Pathogenic fungi are exposed to several host-derived stressors, such as elevated temperature, reactive oxygen species (ROS), reactive nitrogen species (RNS), and deficiencies in essential nutrients. Additionally, morphological switching, a virulence trait in several fungal pathogens, is under significant translational control in *C. albicans* [81] and *Histoplasma capsulatum* [82]. Strikingly, multiple members of the ribosomal subunits (RPS and RPL protein families), ribosome biogenesis factors, and translation regulators have been identified as antigenic proteins in many fungal pathogens, including *A. fumigatus* [52], *C. albicans* [54], *C. glabrata* [55], *C. parapsilosis* [56], *C. posadasii* [60], *C. gattii* [57], *H. capsulatum* [61], *Paracoccidioides* spp. [62,70,71], and *T. marneffei* [64] (Table 3).

The importance of translational control in stress adaptation and viability has been demonstrated in fungal studies. In *C. neoformans*, the disruption of the translation regulators Gcn2 and mRNA deadenylase Ccr4 results in strains (*gcn2*∆ and *ccr4*∆ mutants) that are unable to repress translation and become sensitive to host temperature and oxidative stress [83,84]. Furthermore, the *ccr4*∆ mutant is unable to respond to cell wall stressors and is less virulent in a mouse model of cryptococcosis [85]. Additionally, the chemical inhibition of translation by rocaglates leads to cell death in *Candida auris* [86]. These studies provide information that supports the idea that translation machinery is crucial for host–pathogen interactions and could be potential targets for drug development.

### 4.4. Antioxidant Systems

To detoxify host- and fungal-derived oxidative stress, fungal pathogens have evolved efficient antioxidant systems, including superoxide dismutases, catalases, and the glutathione/glutaredoxin and thioredoxin systems [87]. The significant roles of antioxidant enzymes in fungal fitness and pathogenicity have been clearly elucidated in fungal pathogens [87,88]. Several antioxidant enzymes have been shown to react with human antibodies. First, superoxide dismutase shows immunoreactivity in *C. neoformans* [58]. Second, catalase has been commonly identified as an antigenic protein in *A. fumigatus* [72], *H. capsulatum* [61], *P. brasiliensis* [71], *S. mexicana* [63], and *T. marneffei* [51]. Furthermore, the components of the glutathione/glutaredoxin and thioredoxin systems have been isolated as fungal antigens in *C. albicans* [54], *C. gattii* [57], and *T. marneffei* [64] (Table 3).

## 5. Mechanisms That Render Intracellular Proteins “Antigenic” during Infection: A Perspective

With the current data obtained from immunoproteomic experiments using fungal whole-cell lysates, we now know that intracellular proteins could be targets of HMI. These intracellular fungal antigens can be presented to the adaptive immune system via macrophages, which are known to play a role as antigen-presenting cells [89]. Macrophages can directly ingest and process fungal pathogens or scavenge apoptotic-infected cells and then present the fungal determinants to B-cells and T-cells through the MHCI and MHCII complexes [89]. However, what if there are other ways that antigenic proteins can interact with the host adaptive immune system? As illustrated in Table 3, antigenic proteins function in the chaperone network (i.e., heat-shock proteins), central carbon metabolism, and protein synthesis, which are the known categories of moonlighting proteins [90]. Moonlighting proteins are multifunctional proteins, and switching subcellular localization is one of the mechanisms leading to a change in their functions [90]. Extracellular vesicles (EVs) can be loaded with pathogenesis-related molecules and secreted by fungal pathogens [91,92,93,94,95]. Numerous antigenic proteins have been reported as components of EVs in several fungal pathogens [61,64,95]. With the data collected here, it is tempting to hypothesize that the antigenic proteins are moonlighting proteins that, once translocated through the EV, could switch functions from housekeeping to mediating host interactions. Through EV transportation, these intracellular proteins could be directly exposed to the host immune cells and stimulate antibody production. More experiments would be needed to fill in this knowledge gap.

Therapeutic and vaccination strategies targeting conserved pathways among eukaryotes (such as protein synthesis, carbon metabolic pathways, and environmental stress response) are traditionally considered to be an unfavorable approach due to concerns of an autoimmune response to similar host antigens. However, several experiments have shown that heat-shock proteins and carbon metabolic proteins could stimulate immunity against fungal infections in animal studies (see Section 6 for details [96,97,98]). In addition to the studies mentioned above, the studies on the identification of antigenic proteins provide further evidence that HMI has a protective impact on fungal infections. Specifically, it is noticeable that many antigenic proteins from medically important fungi (such as *H. capsulatum*, *T. marneffei*, and *C. neoformans*) could react with the antibodies present in the sera of both healthy individuals and people with fungal diseases [59,61,68,99]. These antigenic proteins are often excluded from follow-up studies because they are not disease-specific molecules. However, the low incidence of fungal disease in healthy people with the presence of non-disease-specific antibodies could be alternatively interpreted as fungal intracellular proteins having the ability to stimulate HMI, which eventually contributes to the natural resistance of the host to fungi. Thus, it is possible to develop therapeutic strategies targeting this type of antigenic protein.

## 6. Applications of Fungal Antigenic Proteins: Current Progress and Challenges

The identified antigenic proteins have been exploited to generate therapeutic applications in three main areas, including diagnostic tools, monoclonal-based drugs, and vaccines. First, several antigenic proteins have been selected and developed into diagnostic tools that can discriminate patients with different fungal infections or different stages of mycotic disease. In a *T. marneffei* study, the antigenic mannoprotein MP1 was shown to play an important role in virulence [100]. MP1 binds and sequesters arachidonic acid, a key proinflammatory meditator, allowing *T. marneffei* to evade the host’s innate immune defense [101]. MP1 antigen detection tools have been successfully developed and commercially available for the diagnosis of talaromycosis in several clinical settings [102,103]. In a *C. albicans* study, the detection of the antibody-reactivity patterns between the fungal Eno1 and Pgk1 proteins and the sera from patients with a *Candida* infection could differentiate patients with invasive candidiasis from those with non-invasive candidiasis [54]. Moreover, the unique pattern of IgG antibody reactivity with the fungal Tdh3 protein allows for the differentiation of patients with catheter-associated invasive candidiasis from those with non-invasive candidiasis [54]. In an *A. fumigatus* study, the immunogenic thioredoxin reductase GliT protein showed reactivity with the sera from proven invasive aspergillosis patients and, therefore, could be used as a disease marker for an early diagnosis [53]. Overall, several classes of antigenic proteins are valuable in the development of diagnostic tests.

Second, antigenic proteins can be a starting point for the generation of antibodies with the potential application of antifungal drug development. Heat-shock proteins, for example, are a fungal protein class that has been the focus of drug targets for monoclonal-based therapy. Efungumab, also known as mycograb, was the first antifungal drug based on monoclonal antibodies that reached a phase III clinical trial. It was a recombinant fragment of a human monoclonal antibody against *C. albicans* Hsp90 [96]. Treatment with the Hsp90 recombinant antibody significantly improved the survival rates and fungal clearance in animal models of invasive candidiasis [96,97]. In a double-blinded, randomized study conducted by Pachl et al., 2006, a combination treatment between mycograb and lipid-associated amphotericin B significantly reduced the mortality rate associated with *Candida* species in patients with invasive candidiasis [104]. However, the commercial production of mycograb was discontinued due to concerns about the heterogeneity of the recombinant fragments during the production and purification process [105]. In a *Paracoccidioides* spp. study, antibodies against the *P. brasiliensis* and *P. lutzii* Hsp90 proteins were generated in mice. These antibodies successfully bound and opsonized *Paracoccidioides* spp. yeast cells, supporting the idea that Hsp90 is a target for antibodies with therapeutic potential [106]. In a *H. capsulatum* study, monoclonal antibodies against the Hsp60 protein were generated [107]. The survival of mice infected with *H. capsulatum* was significantly prolonged when the Hsp60 monoclonal antibodies were administered to the animals. Notably, the monoclonal antibodies generated by Moura ÁND et al., 2020 and Guimarães AJ et al., 2009 did not recognize mammalian proteins, suggesting that cross-reactivity was likely not a problem. These studies exemplify that the generation of monoclonal antibodies against antigenic proteins can be a promising strategy for managing fungal infections.

Third, antigenic proteins can serve as candidates for vaccine generation. Even though vaccines for invasive mycoses are not currently available, three vaccines have undergone human trials. The first one is PEV7, which is the recombinant aspartyl-proteinase 2 (Sap2) (ClinicalTrials.gov identifier: NCT01067131). The second and third vaccine trials are NDV-3 and its derivatives, containing the recombinant N-terminus of *C. albicans* agglutinin-like sequence 3-protein (Als3p) (ClinicalTrials.gov identifier: NCT01273922 and ClinicalTrials.gov identifier: NCT01926028). These vaccines have been developed and are being evaluated for safety and efficacy against recurrent candida infections. Furthermore, many animal studies have proven the protective role of antigenic proteins against fungal infections. For example, immunized BALB/c mice with the recombinant mannoprotein MP1 could protect against *T. marneffei* infection with 100% survival [108]. Moreover, vaccination with recombinant aspartyl protease Pep1 induced protective immunity against pulmonary coccidioidomycosis in mice [60]. Likewise, vaccination with recombinant Hsp60 could protect mice from pulmonary histoplasmosis and paracoccidioidomycosis [109,110].

There are still many challenges that remain with the development of antifungal drugs and vaccines using antigenic proteins. Due to the similarities between fungi and animal kingdoms, the selection of targets that are specific to fungal pathogens, while not stimulating an immune response in humans, is always a concern. The use of protein antigens that exist only in fungi (such as cell wall components) and/or the selection of fungal epitopes with no homology with human proteins (such as conserved proteins in common metabolic pathways) are potential solutions. Autoimmune response must be carefully evaluated in follow-up experiments and human trials. Another obstacle comes from the economics in developing such drugs and vaccine technologies. Governmental and non-governmental fundings in these projects are limited. With the high cost for research and development of therapeutic agents along with populations most likely affected by fungal infections having little resources, there is limited financial incentives for pharmaceutical companies to focus efforts on these drugs and vaccines. To overcome these formidable challenges, combining resources and efforts from academic, governmental, and private sectors could improve the prospects in preventing, controlling, and diagnosing fungal diseases.

## 7. Concluding Remarks

Overall, there are at least two lessons learned from studies on antigenic proteins in the omics era. First, identified antigens could provide basic knowledge associated with HMI against pathogenic fungi and host–pathogen interactions. Our analysis revealed that these antigenic proteins could be intracellular proteins involved with fitness attributes, serving a conserved housekeeping role to the cells. Thus, proteins with antigenic properties are not limited to those proteins that are involved with the virulence attributes or secretory pathways. Second, the immunoproteomic analysis of fungal antigens could pave the way for developing antibody-based interventions and vaccinations. Indeed, the generation of protective monoclonal antibodies and vaccines against fungal pathogens are currently being investigated, emphasizing the versatility of these identified antigenic proteins [18,65,111]. The identification of new classes of antigenic proteins could lead to future discoveries. For example, many pathogenic fungi can produce secondary metabolites and mycotoxins with deleterious effects. Antibodies against these metabolites can potentially inhibit their adverse effects. However, investigations on antigenic metabolites and the generation of specific antibodies against them have not been carried out so far. In summary, identifying antigenic proteins will continue to provide a wealth of information to microbiologists as well as pharmaceutical and industrial sectors for designing antifungal drugs, diagnostic tools, prophylaxis, and vaccines.

## Figures and Tables

**Figure 1 ijms-25-03531-f001:**
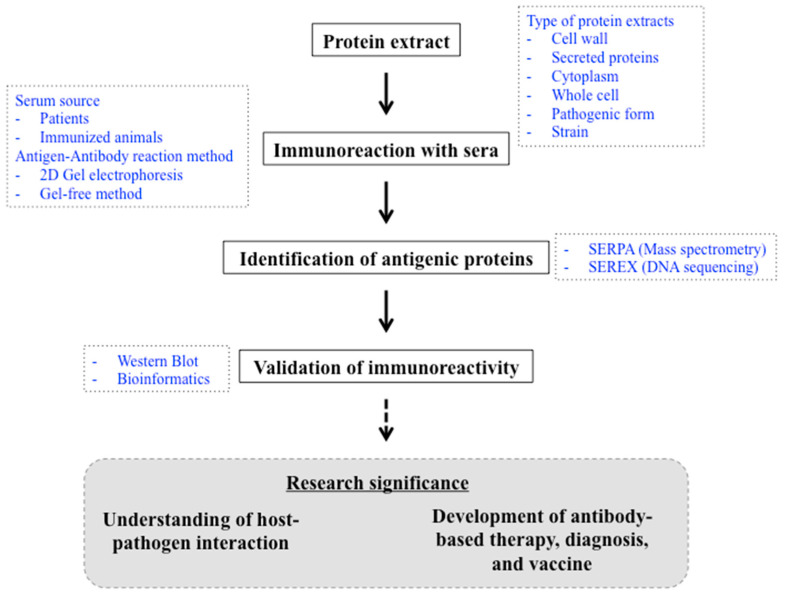
Immunoproteomics is an important approach for the identification of antigenic protein from fungal pathogens. A schematic of the immunoproteomic approach is depicted in the black box. Possible variations in each step are described in blue. The research significance of the identified antigenic protein is described in the grey box.

**Table 1 ijms-25-03531-t001:** Key human inborn errors in B-cell and antibody dysfunctions and the associated mycotic diseases.

Condition	Relevant Effect	Associated Fungal Disease	References
X-linked hyper IgM syndrome	Defective B-cells and elevated IgM	Cryptococcosis	[20,23,25]
		Pneumocystic pneumonia	[22,26,27]
		Candidiasis	[22]
		Histoplasmosis	[22,28]
		Paracoccidioidomycosis	[29]
		Talaromycosis	[21]
CARD9 deficiency	Decreased IgG levels	Candidiasis	[30,31]
Hyper IgE syndrome	Elevated IgE levels	Candidiasis	[32,33,34,35]
		Aspergillosis	[35,36,37]
		Cryptococcosis	[35,38]
		Histoplasmosis	[35,38]
		Coccidioidomycosis	[38]
		Pseudallescheriasis	[38]
		Fusariosis	[35,39]

**Table 2 ijms-25-03531-t002:** Fungal species-specific antigenic proteins.

Species	Method	Key Antigens	Function	References
*Aspergillus fumigatus* (True mold)	- 2D gel analysis of fungal cell wall extracts (germlings) with patient sera + mass spectrometry - 2D gel analysis of fungal extracellular proteins with patient sera + mass spectrometry	- Cytochrome P450 ^D^ - eEF3 ^D^ - Thioredoxin reductase Glit (TR) ^D^	- Xenobiotic detoxification - Translation - Secretory protein	[52,53]
*Candida albicans* (Polymorphic yeast)	- 2D gel analysis of hyphal secretome with patient sera + mass spectrometry	- Bgl2, Eno1, Pgk1, Glx3 ^D^ - Sap5, Pra1, Tdh3 ^D^		[54]
*Candida glabrata* (*Nakaseomyces glabrata*, Yeast)	- Mass spectrometry analysis of secretory proteins + bioinformatics prediction of antigenic proteins - Mice vaccinated with secreted proteins	- Secretome ^V^	- Vaccination with secretome provided protection against infection in mice	[55]
*Candida parapsilosis* (Yeast)	- 2D gel analysis of fungal cell wall extracts with sera from infected mice + mass spectrometry	- Idh2 ^B^	- Isocitrate dehydrogenase (TCA cycle)	[56]
*Cryptococcus gattii* (Yeast) *Cryptococcus neoformans* (Yeast)	- 2D gel analysis of protein extracts from four species with sera from patients + mass spectrometry - 2D gel analysis of cell wall and cytoplasmic proteins with sera from infected mice + mass spectrometry - 2D gel analysis of cell wall and cytoplasmic proteins with sera from patients + mass spectrometry	- Hsp70 ^D^ - GrpE ^D^ - Tpx1 ^D^ - Cell wall and cytoplasmic protein extracts ^V^ - Mpr1 ^V^ - CNAG_02943 ^V^ - HP_06113 ^V^ - UreG ^V^	- Heat-shock protein - Heat-shock protein - Thiol peroxidase - Vaccination with protein fractions provided protection against infection in mice - Extracellular elastolytic metalloprotease - Cytoplasmic protein - Hypothetical protein - Urease accessory protein	[57] [58,59]
*Coccidioides posadasii* (Thermal dimorphic fungi)	- 2D gel analysis of parasitic cell wall extract (spherule) + mass spectrometry	- Pep1 ^V^	Aspartyl protease	[60]
*Histoplasma capsulatum* (Thermal dimorphic fungi)	- Co-immunoprecipitation assay of pathogenic yeast protein extracts with patient sera + mass spectrometry	- YPS3 ^D^ - M antigen ^D^ - Catalase P ^D^	- Yeast-specific cell wall and secreted protein - M antigen - Oxidative stress	[61]
*Paracocidioides* spp. (Thermal dimorphic fungi)	- Immunoprecipitation of exoantigens (secreted antigens) from *Paracoccidioides* spp. with polyclonal antibodies derived from animals immunized with the secretome + mass spectrometry	- PAAG_05807 and PAAG_06925 from *P. lutzii* ^P^	- Hypothetical proteins	[62]
*Sporothrix schenckii* (Thermal dimorphic fungi)	- 2D gel analysis of yeast cell extracts with patient sera + mass spectrometry	- 3-Carboxymuconate cyclase (gp70) ^P^	- Secreted protein	[63]
*Talaromyces marneffei* (Thermal dimorphic fungi)	- Antibody screen using pathogenic yeast recombinant cDNA expression libraries with patient sera + DNA sequencing of positive clones	- P26 ^P^ - Nuo21.3 ^P^ - Nbr1 ^P^		[64]

^D^ = diagnostic values; ^B^ = biomarker; ^P^ = predicted to be species-specific or high immunogenicity/antigenicity; ^V^ = vaccine candidate.

**Table 3 ijms-25-03531-t003:** Common antigenic proteins discovered in pathogenic fungi. The categorization was based on their functions.

Antigenic Proteins	ID	Description	References
**Molecular chaperones** *Aspergillus fumigatus* - Hsp88 - Hsp90/Asp f 12 - Hsp70 - Hsp70 (HscA) - Mitochondrial Hsp70 (Ssc70) - Mitochondrial Hsp60	Q6MYM4 P40292 Q4WJ30 Q4WCM2 Q4X1H5 Q4X1P0	- Heat-shock protein - Heat-shock protein - Heat-shock protein - Heat-shock protein - Heat-shock protein - Heat-shock protein	[52]
*Candida albicans* - Hsp12 - Ssa2 - Hsp90 - Hsp70/Ssa4 - Ssb1 - Ssc1 - Msi3/Sse1		- Hsp12 - Hsp70 Family chaperone - 90-kDa Heat-shock protein - Hsp70 family - Hsp70 family - Hsp70 family - Hsp70 family	[54,69]
*Candida parapsilosis* - Hsp70	P87222	- Heat-shock protein 70 (Ssb1)	[56]
*Coccidioides posadasii* - Hsp70 - Hsp60	DQ674543	- 70-kDa Heat-shock protein - 60-kDa Heat-shock protein	[60]
*Cryptococcus gattii* - Hsp70 - Sks2	CNBG_4912 CNBG_0239	- Heat-shock protein - Heat-shock protein sks2	[57]
*Cryptococcus neoformans* - Heat-shock protein - Hsp90 - Hsp70	AFR94464 AAN76525 AFR97119	- 72-kDa Heat-shock protein - Heat-shock protein 90 - Heat-shock protein 70	[58]
*Paracoccidioides brasiliensis* - Hsp72-like protein - Hsp75-like protein - Hsp60 - Hsp7-like protein *Paracoccidioides lutzii* - Hsp72-like protein - Hsp60 *Paracoccidioides* spp. - Hsp70-like protein - Hsp 60-like protein	PADG_08118 PADG_02761 PADG_08369 PADG_00430 PADG_08118 PADG_08369 PAAG_08003 PABG_05342 PADG_08118 A0A1D2J907 PAAG_08059 PABG_07300 A0A1D2J9F0	- Hsp72-like protein - Hsp75-like protein - Hsp60, Mitochondrial - Hsp7-like protein - Hsp72-like protein - Hsp60, Mitochondrial - 70-kDa Heat-shock protein - 60-kDa Heat-shock protein	[62,70]
*Talaromyces marneffei* - Hsp30	PMAA_014600	- Heat-shock protein 30	[64]
**Carbon metabolism** *Aspergillus fumigatus* - Fructose-bisphosphate aldolase - Enolase (Asp F22) - Aconitase (aconitate hydratase)	Q4WY39 Q96X30 Q4WLN1	- Fructose-bisphosphate aldolase (glycolysis) - Enolase (glycolysis and gluconeogenesis) - Aconitase (TCA cycle)	[52]
*Candida albicans* - Fba1 - Eno1 - TDH3/GAP1 - Tpi1 - Pgk1 - Aco1 - Mdh1		- Fructose-bisphosphate aldolase (glycolysis) - Enolase (glycolysis and gluconeogenesis) - Glyceraldehyde-3-phosphate dehydrogenase (glycolysis) - Triose phosphate isomerase (glycolysis) - Phosphoglycerate kinase (glycolysis) - Aconitase (TCA cycle) - Malate dehydrogenase (TCA cycle)	[54,56,69]
*Candida parapsilosis* - Fba1 - Eno1 - GAP1 - Pgk1 - Idh2	CPAR2_401230 CPAR2_207210 CPAR2_808670 P46273 CPAR2_211610	- Fructose-bisphosphate aldolase (glycolysis) - Enolase (glycolysis and gluconeogenesis) - Glyceraldehyde-3-phosphate dehydrogenase (glycolysis) - Phosphoglycerate kinase (glycolysis) - Isocitrate dehydrogenase (TCA cycle)	[56]
*Coccidioides posadasii* - Aldolase - Enolase - Aconitase - Malate dehydrogenase - NADH-ubiquinone oxidoreductase unit	DQ674539 DQ674538 DQ674544 DQ674541 DQ674550	- Fructose-bisphosphate aldolase (glycolysis) - Enolase (glycolysis and gluconeogenesis) - Aconitase (TCA cycle) - Malate dehydrogenase (TCA cycle) - Electron transport chain (ETC)	[60]
*Cryptococcus gattii* - Enolase (phosphopyruvate hydratase) - Aconitase - GAPDH *Cryptococcus neoformans* - Enolase (phosphopyruvate hydratase)	CNBG_3703 CNBG_0705 CNBG_1866	- Enolase (glycolysis and gluconeogenesis) - Aconitase (TCA cycle) - Glyceraldehyde-3-phosphate dehydrogenase (glycolysis) - Enolase (glycolysis and gluconeogenesis)	[57]
*Histoplasma capsulatum* - Aconitase (aconitate hydratase) - NADH-ubiquinone oxidoreductase 21 kDa unit	C6H4P0 C6HMG0 A6QUB6	- Aconitase (TCA cycle) - Electron transport chain (ETC)	[61]
*Paracoccidioides brasiliensis* - Aldolase - Aconitase - GAPDH - TPI *Paracoccidioides lutzii* - Aldolase - Enolase - GAPDH - TPI - ICL *Paracoccidioides* spp. - MDH - ICL	PADG_00668 PADG_11845 PADG_02411 PADG_06906 PADG_00668 PADG_04059 PADG_02411 PADG_06906 PADG_01483 PAAG_08449 PADG_08054 A0A1E2Y1Z7 PAAG_06951 A0A1E2YBS1	- Fructose-bisphosphate aldolase (glycolysis) - Aconitase (TCA cycle) - Glyceraldehyde-3-phosphate dehydrogenase (glycolysis) - Triose phosphate isomerase (glycolysis) - Fructose-bisphosphate aldolase (glycolysis) - Enolase (glycolysis and gluconeogenesis) - Glyceraldehyde-3-phosphate dehydrogenase (glycolysis) - Triose phosphate isomerase (glycolysis) - Isocitrate lyase (glyoxylate cycle) - Malate dehydrogenase (TCA cycle) - Isocitrate lyase (glyoxylate cycle)	[62,70,71]
*Talaromyces marneffei* - Fbp1 - Nuo21.3	PMAA_041280 PMAA_028280	- Fructose-1,6-bisphosphatase (glycolysis/gluconeogenesis) - NADH-ubiquinone oxidoreductase (ETC)	[64]
**Protein synthesis** *Aspergillus fumigatus* - 60S ribosome biogenesis protein Sqt1 - RL3_NEUCR 60S ribosomal protein L3 - Translation elongation factor eEF-1 subunit gamma - Translation elongation factor eEF-3 - Elongation factor Tu	Q4WU69 Q5AZS8 Q4WDF5 Q4WGN6 Q8TGG6	- Ribosome biogenesis - Ribosomal protein - Protein synthesis - Protein synthesis - Protein synthesis	[52]
*Candida albicans* - Asc1 - Tif		- Ribosomal protein - Translation initiation factor (protein synthesis)	[54]
*Candida glabrata* (*Nakaseomyces glabrata)* - Rps8A - Rps29B - Asc1 - Rpl12B - Rpl26A - Rpl32 - Rps24A - Rps21A - Rpl10 - Rpl33A	CAGL0A04521g CAGL0D00858g CAGL0D02090g CAGL0F02937g CAGL0G01078g CAGL0H04521g CAGL0J03234g CAGL0K08382g CAGL0K12826g CAGL0M02497g	- Ribosomal protein - Ribosomal protein - 40S Ribosomal subunit - Ribosomal protein - Ribosomal protein - Ribosomal protein - Ribosomal protein - Ribosomal protein - Ribosomal protein - Ribosomal protein	[55]
*Candida parapsilosis* - Tif1	P87206	- Eukaryotic initiation factor 4A (translation apparatus)	[56]
*Cryptococcus gattii* - 40S ribosomal protein S0 - 40S ribosomal protein S7 - Initiation factor 5a - Elongation factor 1-beta - Translation elongation factor EF1-alpha	CNBG_2923 CNBG_2617 CNBG_5941 CNBG-3378 CNBG_4834	- Ribosomal protein - Ribosomal protein - Protein synthesis - Protein synthesis - Protein synthesis	[57]
*Histoplasma capsulatum* - Large subunit ribosomal protein l3 - Ribosomal l10 protein - Ribosomal protein l14 - Ribosomal protein l15 - Ribosomal protein l22 - Ribosomal protein l31e - Ribosomal protein l34 protein - Ribosomal protein l37 - Ribosomal protein l37a - Ribosomal protein s13 - Ribosomal protein s16 - Ribosomal protein s2 - Ribosomal protein s5 - Ribosomal protein s9 - Ribosomal protein srp1 - Ribosomal protein yml20	C0NDC6 C0NCP4 C0NHN4 A6R1V3 A6R1J7 A6R6D4 C6H9U0 F0U7R8 F0UB50 A6R4V4 A6R4L7 F0URZ8 A6RE96 A6REK8 C0NLR4 C6HP82	- Translation - Protein synthesis - Protein synthesis - Protein synthesis - Protein synthesis - Protein synthesis - Protein synthesis - Protein synthesis - Protein synthesis - Protein synthesis - Protein synthesis - Protein synthesis - Protein synthesis - Protein synthesis - Protein synthesis - Protein synthesis	[61] See the references for the full list
*Paracoccidioides* spp. - 40S ribosomal protein S15	PAAG_04690 A0A1E2YDH8	- Ribosomal protein	[62]
*Talaromyces marneffei* - RPL20A	PMAA_054240	- 60S Ribosomal protein L20A	[64]
**Antioxidative system** Catalase *Aspergillus fumigatus* - Catalase		- Catalase-peroxidase	[72]
*Histoplasma capsulatum* - Catalase	C0NVF6 Q9Y7C2	- Catalase	[61]
*Paracoccidioides brasiliensis* - Catalase			[71]
*Sporothrix mexicana* - Catalase/peroxidase	SS08703/SB01256	- Catalase-peroxidase	[63]
*Talaromyces marneffei* - CpeA	Q8NJN2.1	- Catalase-peroxidase	[51]
Glutathione system *Candida albicans* - Prx1		- Thioredoxin peroxidase	[54]
*Cryptococcus gattii* - Glutathione transferase - Tpx1 - Grx5	CNBG_6043 CNBG_2132 CNBG_5485	- Xenobiotic detoxification - Thioredoxin peroxidase - Glutaredoxin Grx5-prov protein	[57]
*Talaromyces marneffei* - Gpx1	PMAA_007230	- Glutathione peroxidase	[64]
*Cryptococcus neoformans* - Superoxide dismutase	AFR97119	- Mitochondrial superoxide dismutase	[58]

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
