# Peer review of "Human–Fungal Pathogen Interactions from the Perspective of Immunoproteomics Analyses"

_ijms, 2024, doi:10.3390/ijms25063531_

Round 1
Reviewer 1 Report
Comments and Suggestions for Authors
The authors have prepared a comprehensive review of the relevant data supporting the human-fungal pathogen interactions. This review is a good contribution to the field and summarizes a large set of studies concisely and clearly. I have included only a few comments.
1. The manuscript does a nice job describing the comprehensive set of potential human-fungal interactions that could be exploited to generate new therapeutics, vaccines or diagnostics. There is reference to a few that have been tried, but no specific section on these attempts and the successes and failures. Did the authors consider if adding a bit more detail on this?
2. Line 187, it states "significance of identified antigenic proteins in the legend; however, should this be the proteins identified or the method to identify the proteins, based on the information in the figure?
Reviewer 2 Report
Comments and Suggestions for Authors
This review provided comprehensive information regarding to antigenic proteins in medically important fungi for preventive, vaccination and therapeutic strategies against fungal emerging infections.Bellow, some comments might help to increase the importance of this study.
1.Considering antibodies to fight antigenic proteins also might react some human antigens as a cross reaction, how do you explain the therapeutic effect of those antibody to treat fungal infections?
2.Are there in vivo(animal models)used some of antigenic proteins of fungal species to eliminate fungal infections in review literature with specific function?If soothe authors should include this part in the review.
3.Table 3,antigenic proteins list for H.capsulatum is too long list, it would be more appealing to summarise it only the main antigens and for others just mention the relevant references in the text.
4.the authors illuminated many antigens for different fungal species ,it is recommended for each species importantly focused on the most substantial ones based on the literature, especially for those included in invite mice models with therapeutic approach to eradicate fungal load from infected tissue.
Comments on the Quality of English Language
To improve the English quality of paper, edition by Native English speaker is recommended throughout the text.
